# Comparison of Waterlogging Tolerance of Three Peach Rootstock Seedlings Based on Physiological, Anatomical and Ultra-Structural Changes

**Fangjie Xu** [1,†]  , **Huaqing Cai** [2,†], **Xianan Zhang** [1], **Mingshen Su** [1], **Huijuan Zhou** [1], **Xiongwei Li** [1], **Minghao Zhang** [1], **Yang Hu** [1], **Chao Gu** [2], **Jihong Du** [1,*] and **Zhengwen Ye** [1]

1   Shanghai Key Lab of Protected Horticultural Technology/Forestry and Pomology Research Institute, Shanghai Academy of Agricultural Sciences, No. 1000 Jinqi Road, Fengxian District, Shanghai 201403, China
2   Centre of Pear Engineering Technology Research, State Key Laboratory of Crop Genetics and Germplasm Enhancement, Nanjing Agricultural University, Nanjing 210095, China
*   Correspondence: dujihong@saas.sh.cn; Tel.: +86-021-37195610
†   These authors contributed equally to this work.

**Abstract:** Peach (*Prunus persica* (L.) Batsch) is a typical shallow-rooted fruit plant with a high respiratory intensity and oxygen demand, which makes it highly susceptible to oxygen-deficient soil conditions resulting from waterlogging. Rootstock waterlogging resistance is essential to the performance of cultivated peaches under waterlogging stress. In comparison to *Prunus persica* var. *persica* ('Maotao', M) and *Prunus davidiana* (Carr.) C. de Vos ('Shantao', S), *Prunus persica* f. Hossu ('Hossu', H) exhibited superior leaf photosynthetic electron transfer efficiency, a higher rate of mycorrhizal fungi infection in both fine roots and mesophyll palisade cells, as well as earlier air cavity formation in both leaf midvein and fine roots under waterlogging stress. Furthermore, under non-waterlogging conditions, Hossu had greater leaf superoxide dismutase (SOD) activity, higher proline content, and a greater content of starch granules in the pith and xylem ray cells of stems and roots than rootstocks M and S. As a result, Hossu's tolerance to waterlogging may be due to its higher photosynthetic efficiency, improved tissue oxygen permeability, higher energy metabolism, and increased intracellular mycorrhizal fungus infection rates in both root parenchyma cells and mesophyll palisade cells.

**Keywords:** waterlogging; peach rootstocks; photosynthetic responses; antioxidative and osmotic regulation; anatomical adaptation; mycorrhizal fungus infection rates

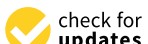



## 1. Introduction

With global climate change and ecosystem destruction, rainstorm and flood disasters have become more common in some areas, combined with improper irrigation and poor soil drainage, causing waterlogging damage to plants [1–5]. Excess/seasonal concentrated precipitation is common in peach-producing regions of southern China during the peach fruit development period, which is usually accompanied by waterlogging. Peach (*Prunus persica* (L.) Batsch) is a typical shallow-rooted fruit plant with high respiratory intensity and oxygen demand, and it is one of the most vulnerable to waterlogging-induced hypoxia among Rosaceae fruit trees [4,6,7]. Consequently, waterlogging has become one of the major abiotic stresses limiting peach fruit yield and quality [1–3,8].

Soil hypoxia occurs within days or even hours after waterlogging, and is a critical factor impacting plant growth and development or causing serious physiological damage [2,9,10]. It adversely affects the root function and consequently the plant growth, with a potential to cause leaf chlorosis, leaf and fruit drop, or even plant death [11–13]. Therefore, the waterlogging tolerance of rootstocks is the most critical factor affecting the overall performance of fruit trees under waterlogging stress [5,14–18]. It is one of the important pathways for

enhancing the waterlogging tolerance of cultivated peach by thoroughly studying the water-logging tolerance mechanism of peach rootstocks and breeding peach rootstocks with higher waterlogging tolerance [6,19,20].

Until recently, however, the majority of the available research on waterlogging tolerance mechanisms and breeding of waterlogging tolerant germplasm focused on cash crops [15,21], field crops [22–29], or vegetable plants [30]. There are very few reports on the internal correlation between the anatomical/ultra-structural changes of various plant tissues and their effects on the waterlogging tolerance performance of peach trees, despite the fact that the majority of the previous research on waterlogged peach trees focused on photosynthetic physiological mechanisms [4,5,31], changes in root architecture [32], osmotic regulation mechanisms [20], and antioxidant responses [4]. Additionally, the breeding of peach rootstocks with great resistance to waterlogging is yet in its infancy. Accordingly, two conventionally used commercial peach rootstocks, *Prunus persica* var. *persica* ('Maotao', M) with relatively well-performing in waterlogging tolerance and *Prunus davidiana* (Carr.) C. de Vos ('Shantao', S) with relatively poor waterlogging tolerance were used as the control (CK) in the present study. The response mechanisms of a new type of peach rootstock, 'Hossu' (H) to short-term waterlogging, including morphological and physiological changes, as well as anatomical and ultrastructural responses, was comprehensively investigated, as was H's waterlogging tolerance.

## 2. Materials and Methods

### 2.1. Plant Material Preparation and Growth Conditions

Six-month-old seedlings of Prunus persica var. persica ('Maotao', M), Prunus davidiana (Carr.) C. de Vos ('Shantao', S) and Prunus persica f. Hossu (H) served as materials in this study. M and S were used as the controls (with varied waterlogging tolerance). These seedlings were placed in plastic pots (18 cm in diameter × 18 cm in height) filled with the mixed substrate (peat: vermiculite: perlite = 1:1:1, *v/v/v*) and well-watered with total nutrient solution before the start of the experiment.

The experiment was conducted from July to August of 2021 in a greenhouse under natural lighting conditions, which was located in Fengpu Campus, Shanghai Academy of Agricultural Sciences in Shanghai, China (116°41′33″ N, 39°91′09″ E, annual average temperature 15.8 °C, annual precipitation 1161.1 mm). Temperature settings in the greenhouse were 28 °C/20 °C (±5 °C, day/night), with a relative humidity of about 70 ± 5%.

### 2.2. Experimental Design and Treatments

During the 14 days of waterlogging treatment (WT), the seedlings do not require any irrigation. However, before the waterlogging treatment began, the tested seedlings were thoroughly watered once every three days. During the WT, the potted seedlings were placed into a plastic container measuring 61 cm long, 42 cm wide, and 26 cm high. Tap water was then injected into the container until the liquid level was flush with 3–4 cm above the substrate surface. The WT lasted for 14 days, and a randomized block design was used. Each treatment had four replicates, with six individuals (seedlings) in each replicate. The non-stressed condition (M-0, S-0, H-0, the relative water content (RWC) of the substrate is 60%), the first (M-1, S-1, H-1), third (M-3, S-3, H-3), seventh (M-7, S-7, H-7), and fourteenth (M-14, S-14, H-14) day of WT were chosen as sampling times, respectively.

### 2.3. Plant Vegetative Growth Measurements

The plant height (PH) of seedlings refers to the distance between the top growth point of shoots to the base of the main stem, and it was determined using a ruler.

The stem diameter (SD) was measured at 4 cm above the base of the main stem, using an electronic digital caliper.

The relative chlorophyll content (soil plant analysis development (SPAD) values) was assessed on fully-matured leaves in the middle part of the seedlings using a chlorophyll meter (SPAD-502 Plus, Minolta Camera Co., Ltd., Osaka, Japan).

## 2.4. Observation of Stomatal Morphology

The modified nail polish smear method [33] was used to observed the stomatal morphology of fully developed, matured leaves located in the upper part of plants. Transparent nail polish was applied evenly to the surface of the leaf's lower epidermis, the dried nail polish film was transferred onto the glass slide moistened with distilled water, it was dried at 37 °C, and then examined under a microscope (Nikon Eclipse E200MV RS NIKON Corporation, Tokyo, Japan). To determine the stomatal width (SW) and stomatal length (SL) of 20 randomly selected stomata, photographs of each leaf sample were recorded by ImageJ software (ImageJ version 1.50i, Wayne Rusband, National Institute of Health, Bethesda, MA, USA).

## 2.5. Photosynthetic Parameters Measurements

The CIRAS-3 portable photosynthesis system (CIRAS-3, PP System, Amesbury, MA, USA) with a 1.75 cm$^2$ leaf chamber was used to measure the photosynthetic parameters. The intensity of the built-in red and blue light source was 2000 µmol m$^{-2}$ s$^{-1}$, the instrument leaf chamber temperature was 25 °C, the air relative humidity was 100%, and the leaf chamber $CO_2$ flow rate (Cs) settled in this study was 390 µmol s$^{-1}$. Measurements were conducted on fully matured leaves from 9:00 to 11:00 a.m. on sunny days. At 0, 1st, 3rd, 7th, and 14th day of WT, the net photosynthetic rate (Pn), intracellular $CO_2$ concentration (Ci), stomatal conductance (gs), transpiration rate (E), and water use efficiency (WUE) were measured. The stomatal limitation value (Ls), which was determined using the formula: Ls = 1 − Ci/Cs, was then calculated.

## 2.6. Determination of Chlorophyll Fluorescence Parameters

Using a portable fluorometer Handy-PEA (Hansatech, Kings Lynn, Norfolk, UK), chlorophyll fluorescence was also recorded at 0, 1st, 3rd, 7th, and 14th day of WT. The analysis was carried out after treatment with 2000 µmol m$^{-2}$ s$^{-1}$ light intensity following 20 min of dark adaptation.

The following chlorophyll indexes were measured in this study: Fo (minimum chlorophyll a fluorescence), Fm (maximum chlorophyll a fluorescence after 20 min of dark adaptation), Fv/Fm (The maximum quantum yield of photosystem II (PSII) photochemistry), Fv/Fo (ratio of the photochemical and non-photochemical processes in PSII), ABS/RC (absorbed photon flux per PSII reaction center (RC)), ETo/RC (the flux of electrons transferred from the primary electron acceptor per active PSII RC), DIo/RC (the flux of energy dissipated in processes other than trapping per active PSII RC), and TRo/RC (the flux of electrons trapped from electron transportation chain per PSII RC). All measurements were carried out on matured leaves randomly [23,34].

## 2.7. Determination of Antioxidant Response System

### 2.7.1. Preparation of Crude Enzyme Extract

The excised fresh leaf samples were frozen with liquid nitrogen and homogenized with 100 mM phosphate-buffered saline (PBS, pH = 7.4, ThermoFisher Scientific, Waltham, MA, USA). The mixture was centrifuged at 12,000× *g* for 20 min at 4 °C and the supernatant was used as the crude extract for the subsequent analyses.

### 2.7.2. Determination of Antioxidant Enzyme Activities, Malondialdehyde, and Proline Content

Different colorimetry kits purchased from Nanjing Jiancheng Bioengineering Institute (Nanjing, China) were used to measure the activities of superoxide dismutase (SOD, EC 1.15.1.1), catalase (CAT, EC 1.11.1.6), peroxidase (POD, EC 1.11.1.7), as well as the content of malonaldehyde (MDA) and proline. The detailed test procedures refer to the kits' instructions.

### 2.8. Determination of Root Vigor and Root Respiration Intensity

Using the TTC detection kit purchased from Nanjing Jiancheng Bioengineering Research Institute (Nanjing, China), the root vigor was assessed using the 2, 3, 5-triphenyltetrazolium chloride (TTC) reduction method.

By using Oxytherm (Hansatech Instrumental, Norfolk, England), the respiration rate of roots was quantified and expressed in nanomoles of oxygen consumed per gram of root (fresh weight) per minute (nmol g $FW^{-1}$ $min^{-1}$).

### 2.9. Observation of Anatomical and Ultra-Structure of Roots, Stems and Leaves

By cutting sections of paraffin-wax, the anatomical structure of roots, stems, and leaves was examined. The primary lateral roots (5–10 mm segments), fine roots (5–10 mm of root tips), the basal part of the stem, and fully matured leaves were collected respectively and immersed in FAA fixative (mixture consisting of 38% (*v/v*) formalin solution, 70% (*v/v*) alcohol, and glacial acetic acid (5: 90: 5, *v/v/v*)) for at least 24 h. Following hydration with a succession of graded alcohol and transparency with Histo Clear solution (National Diagnostics, Atlanta, GA, USA), the samples were embedded in molten paraffin at 58 °C, cooled to room temperature, and then cut it into 8 μm thick slices along a specific direction using a microtome (Leica RM2265, Leica Biosystems Nussloch GmbH D-69226 Nussloch, Nußloch, Germany). After toluidine blue (TB) staining, the samples were examined under a microscope (Nikon Eclipse E200MV RS NIKON Corporation, Tokyo, Japan). All micro-instance measurements were carried out by ImageJ software (ImageJ version 1.50i, Wayne Rusband, National Institute of Health, Bethesda, MA, USA).

The epidermal cortex layer, phloem, xylem, and pith of primary lateral roots and stems were examined using a scanning electron microscope (SEM, TM4000Plus, Hitachi, High-Technologies, Tokyo, Japan) according to the method described by Sánchez-Mata et al. [35].

The mesophyll tissues of leaves and fine roots were cut into pieces measuring 1 $mm^2$ and 3 mm-long segments, respectively, for the transmission electron microscope (TEM, JEM-1010, JEOL Co., Ltd., Tokyo, Japan) observation. These samples were immediately soaked in 100 mM phosphate buffer (PB, pH 7.2, ThermoFisher Scientific, Waltham, MA, USA) for 40 min, followed by 1% (*w/v*) osmic acid for 2 h. Using an ultramicrotome (Thcnai G2, ThermoFisher, Waltham, MA, USA), ultrathin sections (100 nm thick) were created after dehydration in a gradient series of acetone, embedding in Epon 812, and polymerization at 60 °C for 48 h. The slices were then examined at an accelerating voltage of 200 kV after being stained with 0.5% (*w/v*) uranium acetate and 1% (*w/v*) lead citrate.

### 2.10. Statistical Analysis

The data were expressed as the mean ± standard deviation (SD) with at least six replicates chosen for all biochemical analyses. Statistical analysis was performed by SPSS 17.0 (SPSS, Chicago, IL, USA). The significant differences ($p < 0.05$) between the means were determined using Duncan's multiple range comparison tests. Graphing was performed using GraphPad Prism 6 (Lo Jolla, CA, USA).

## 3. Results

### 3.1. Effect of Waterlogging on Plant Height, Stem Diameter and Leaf SPAD Values

As illustrated in Figure 1, waterlogging treatment (WT) for 14 days had no discernible effect on the plant height and stem diameter of any seedlings evaluated, despite significant differences in the change patterns of leaf SPAD value. With the prolongation of WT, the SPAD value of top leaves of 'Maotao' (M) declined continually while that of 'Shantao' (S) remained relatively high and consistent. The SPAD value of lower leaves of S increased transiently on the third day of WT and returned to a level comparable to the very beginning; that of M temporarily increased on the third day of WT and then performed a temporary decline and then rebounded during the early stage of WT (WT 1–3 d). In contrast, throughout 14 days of WT, the SPAD values of the 'Hossu' (H) plant's upper and lower leaves

remained higher and steady. This indicated that during the waterlogging, H had higher leaf chlorophyll content than M and S.

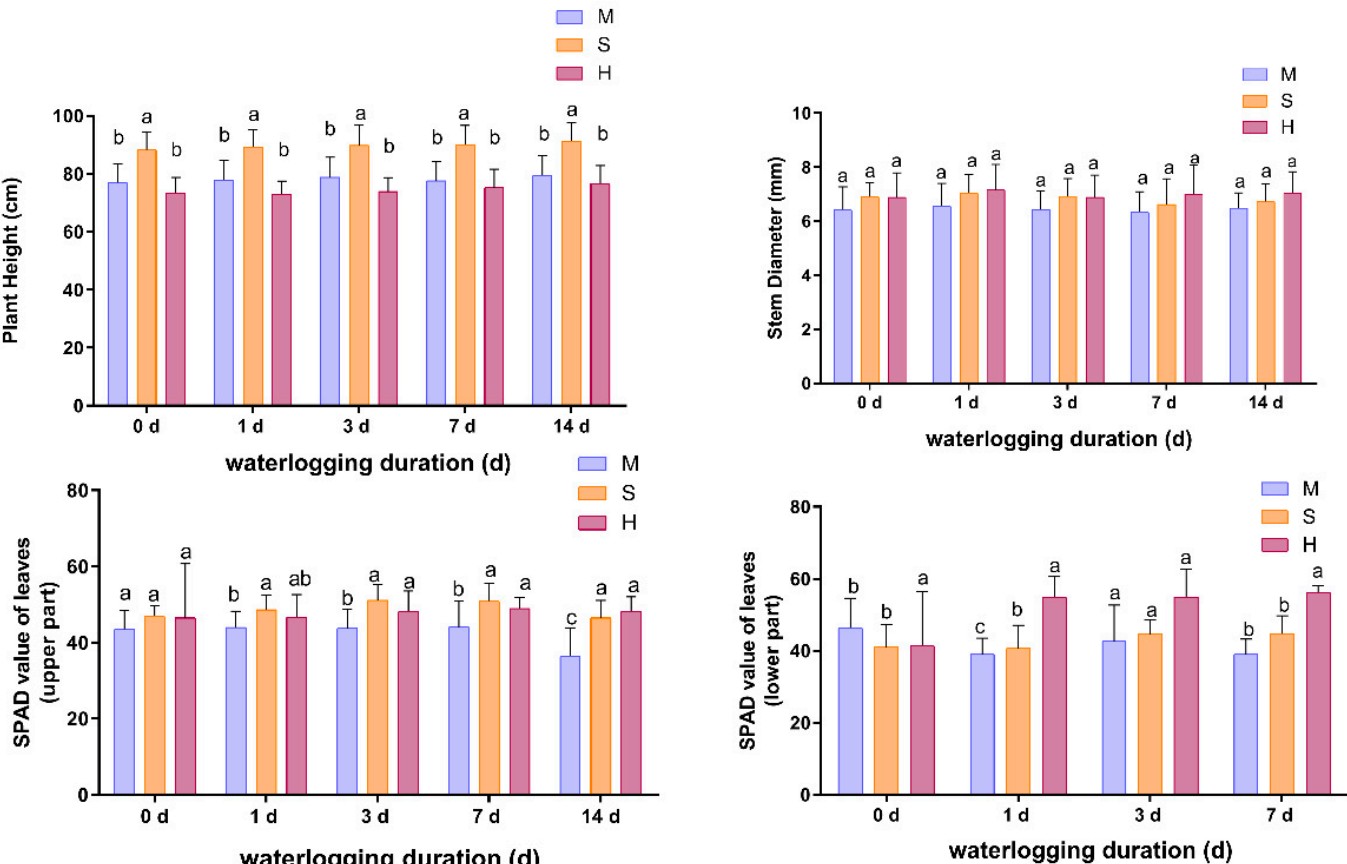

**Figure 1.** The changes of plant height, the stem diameter and leaf SPAD value of three peach rootstocks seedlings with the prolongation of waterlogging exposure. Abbreviations: M: Prunus persica, 'Maotao'; S: Prunus davidiana, 'Shantao'; H: 'Hossu'. The data are the means ± SDs. Within columns, means with a different letter indicate significant differences ($p < 0.05$) using Duncan's test, bars on the columns represent the standard error of the mean (n = 6). The defoliation of S was too severe to provide enough samples for SPAD value measurements of lower part leaves at the 14th day of WT, therefore the leaf SPAD values on the 14th day of waterlogging are not shown in Figure 1.

### 3.2. Effect of Waterlogging on Leaf Stomatal Density and Morphology

As shown in Table 1, the pre-waterlogged leaf stomatal density of S was much lower than that of M and H. The stomatal density of M and S increased modestly during the early stage of WT (1–3 d) but decreased quickly with the prolonged waterlogging exposure, while that of H showed an opposite tendency in that it decreased at first and then increased. Until the end of the waterlogging treatment (WT-14 d), the leaf stomatal density of H was even larger than that of the non-stressed condition, and M displayed a higher stomatal closure ratio (%) than the other two rootstocks. After 14 days of WT, the leaf stomatal morphology of three peach rootstocks was compared in Figure 2.

**Table 1.** Comparison of leaf stomatal density and stomatal closure ratio of different peach rootstocks under waterlogging stress.

| Rootstocks | Stomatal Density (Numbers/mm$^2$) | | | Stomatal Closure Ratio (%) | | |
|---|---|---|---|---|---|---|
| | WT-0 d | WT-3 d | WT-14 d | WT-0 d | WT-3 d | WT-14 d |
| M | 286.61 ± 24.61 ab | 301.25 ± 7.62 a | 216.33 ± 5.69 c | 7.70% | 21.90% | 67.20% |
| S | 205.62 ± 23.65 cd | 215.33 ± 12.03 c | 203.58 ± 5.68 cd | 14.50% | 36.00% | 52.20% |
| H | 275.74 ± 13.21 b | 205.62 ± 17.56 cd | 306.35 ± 8.19 a | 10.20% | 36.20% | 56.30% |

Abbreviations: M: Prunus persica, 'Maotao'; S: Prunus davidiana, 'Shantao'; H: 'Hossu'. The data are the means ± SDs. A different letter indicates significant differences ($p < 0.05$) using Duncan's test (n = 6).

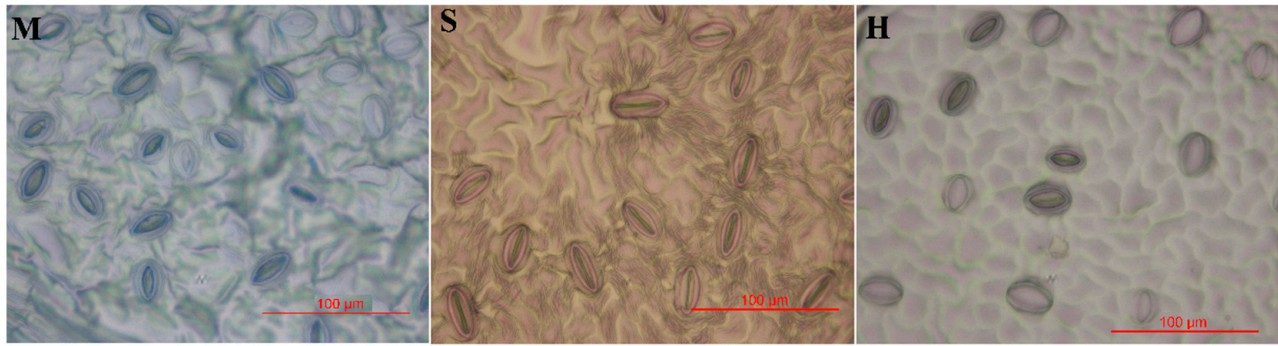

**Figure 2.** Leaf stomatal morphology of three peach rootstocks after 14 d of waterlogging (Bar = 100 μm). Abbreviations: M: Prunus persica, 'Maotao'; S: Prunus davidiana, 'Shantao'; H: 'Hossu'.

### 3.3. Effect of Waterlogging on Leaf Stomatal Morphological Parameters

As shown in Table 2, although the stomatal width of M also decreased, its stomatal length increased significantly at the end (WT-14 d), while the stomatal width and stomatal length of both S and H displayed a comparable, continuously decreasing trend with the prolongation of waterlogging exposure.

**Table 2.** Comparison of leaf stomatal morphological changes of different peach rootstocks under waterlogging stress.

| Cultivar | Waterlogged Duration (d) | Stomatal Width (μm) | Stomatal Length (μm) |
|---|---|---|---|
| M | WT-0 d | 6.41 ± 1.42 a | 16.59 ± 2.20 b |
| | WT-3 d | 6.54 ± 0.56 a | 15.46 ± 1.49 b |
| | WT-14 d | 5.74 ± 0.93 b | 21.90 ± 1.80 a |
| S | WT-0 d | 8.03 ± 0.75 a | 23.96 ± 1.52 a |
| | WT-3 d | 5.95 ± 1.13 b | 19.79 ± 1.69 b |
| | WT-14 d | 3.89 ± 0.52 c | 19.00 ± 1.39 b |
| H | WT-0 d | 6.45 ± 1.20 a | 19.26 ± 1.97 ab |
| | WT-3 d | 5.78 ± 0.73 b | 19.81 ± 0.91 a |
| | WT-14 d | 3.26 ± 0.69 c | 12.79 ± 1.10 c |

Abbreviations: M: Prunus persica, 'Maotao', S: Prunus davidiana, 'Shantao', H: 'Hossu'. The data are the means ± SDs. The different letters indicate significant differences ($p < 0.05$) using Duncan's test (n = 6).

### 3.4. Effect of Waterlogging on the Photosynthetic Capacity

As shown in Figure 3, the net photosynthetic rate (*Pn*) and stomatal conductance (*gs*) of all three rootstocks decreased with prolonged waterlogging exposure. S performed a significantly greater decrease in *gs* and *Pn* (91.7% and 85.3%, respectively) at the initial stage (0–1 d) of WT compared to M (54.5% and 34.5%, respectively) and H (59.9% and 36.1%, respectively), which may indicate that the leaf photosynthetic system of S was the most sensitive to waterlogging.

The transpiration rate (*E*) of S and H initially decreased (WT 0–1 d) and then briefly increased (WT-3 d) before declining again. In contrast, the E value of M declined steadily throughout WT. At the beginning of WT (1–3 d), the intracellular $CO_2$ concentration (*Ci*) and *Pn* of M and H displayed a similar tendency, but with the prolonged waterlogging exposure, the trend changed. This finding confirmed that non-stoma limitation factors began to predominately regulate *Pn* value during the later stage of waterlogging.

At the initial stage of WT (WT 0–1 d), M performed a brief increase in *WUE* value. During the early stage of waterlogging (WT 1–3 d), M maintained a comparatively high level before gradually declining until the end of WT. The WUE value of S decreased continuously and even reached a negative value on the 7th day of WT, which may indicate that the photosynthetic properties of S have been irreversibly damaged after 7 days of waterlogging. In contrast, *WUE* value of H maintained a high and stable level during the first three days before decreasing steadily.

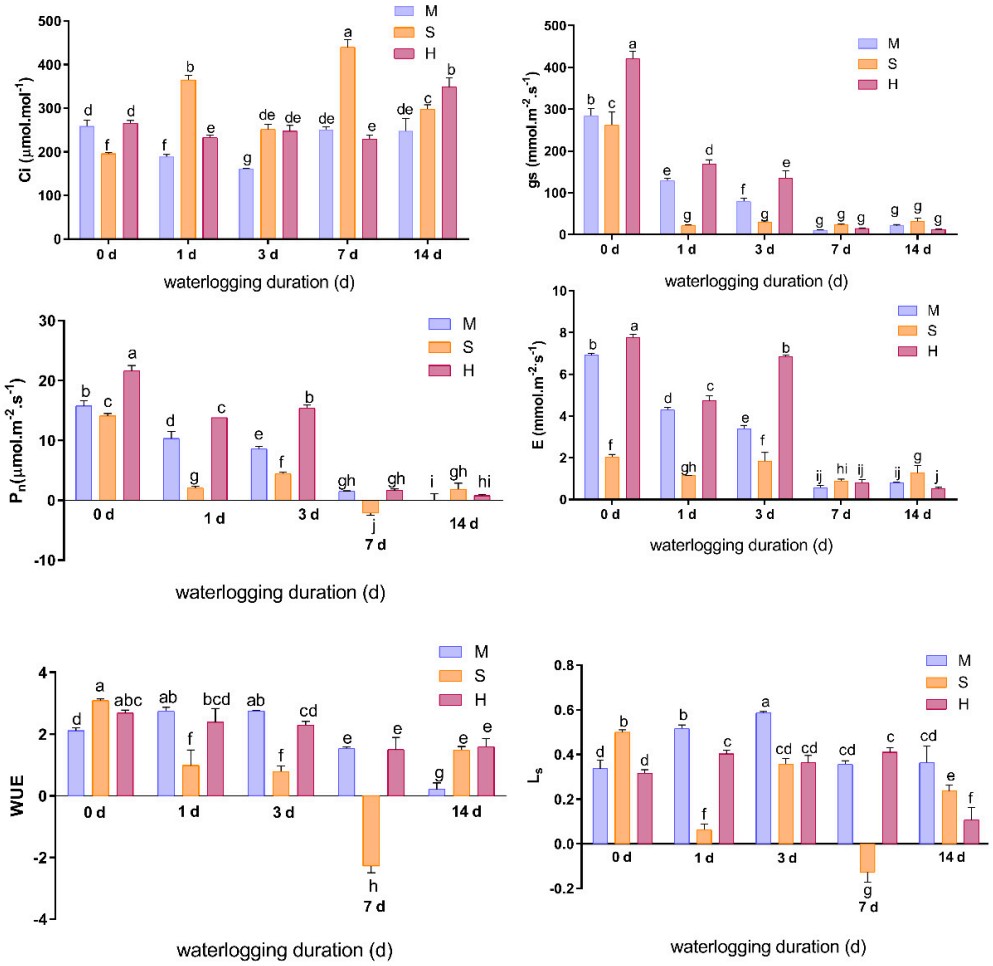

**Figure 3.** The photosynthetic parameters of leaves of different peach rootstocks under waterlogging treatment. Abbreviations: M: Prunus persica, 'Maotao'; S: Prunus davidiana, 'Shantao'; H: 'Hossu'. The data are the means ± SDs. Within columns, means with a different letter indicate significant differences (*p* < 0.05) using Duncan's test; bars on the columns represent the standard errors (SD) (n = 6).

### 3.5. Effects of Waterlogging on Chlorophyll Indexes

According to Figure 4, the value of Fv/Fo and Fv/Fm first increased briefly, then steadily declined until the end of WT (WT-14 d); M had a greater decrease in the value of Fv/Fm value (15.1%) than that of S (7.1%) and H (9.8%). The Fv/Fo value of M decreased by 43.3%, which was more than that of S (31.1%) and H (37.5%), respectively.

The Fo value initially decreased (WT 0–1 d) before gradually increasing. M and H showed temporary declines in Fo value of 22.0% and 10% at first (WT 0–1 d), but eventually bounced back and increased by 27.3% and 45.8% compared to the non-stressed condition, respectively. In contrast, S first declined by 22.5% before recovering slightly and displaying a decrease of 6.8% during the whole 14 days. This indicated the chlorophyll content of S decreased while that of M and H increased under waterlogging.

The principles of light energy absorption, transformation, and utilization in different peach rootstocks seedlings under waterlogging stress were shown in Figure 5. The value of ABS/RC in M and S initially decreased momentarily (WT 0–1 d and WT 1–3 d, respectively), recovered, and remained at a relatively high level during the later stage of WT (WT 7–14 d), whereas that of H significantly increased at first (WT-1 d) and then sustained at a high level throughout the 14 d of WT. The total increment of ABS/RC of M, S, and H was 21.3%, 38.2%, and 39.5%, respectively. The values of TRo/RC and ABS/RC performed a similar changing pattern, whereas the value of ETo/RC varied: the ETo/RC of S decreased constantly and that of M increased initially before decreasing dramatically until the end. In contrast, the ETo/RC value of H was maintained at a relatively consistent and high level throughout the overall 14 d of WT, which might indicate that H maintained better light energy usage efficiency than that of M and S under the same level of waterlogging stress.

The dissipated heat that the leaf PSII reaction center (RC) cannot use was indicated by the value of DIo/RC. It is possible to draw the conclusion that at the late stage of WT, the relatively higher value DIo/RC of M represents excess light energy absorbed by its leaf PSII RC that was immediately dissipated in the form of heat under waterlogging. The DIo/RC value of H increased continuously and slightly with the prolongation of waterlogging, and the increase during the later stage of waterlogging (WT 7–14 d) was greater than that in the earlier stage (WT 0–7 d). In contrast, S suffered a brief decrease in heat dissipation at first (WT 0–3 d) before rebounding and increasing later.

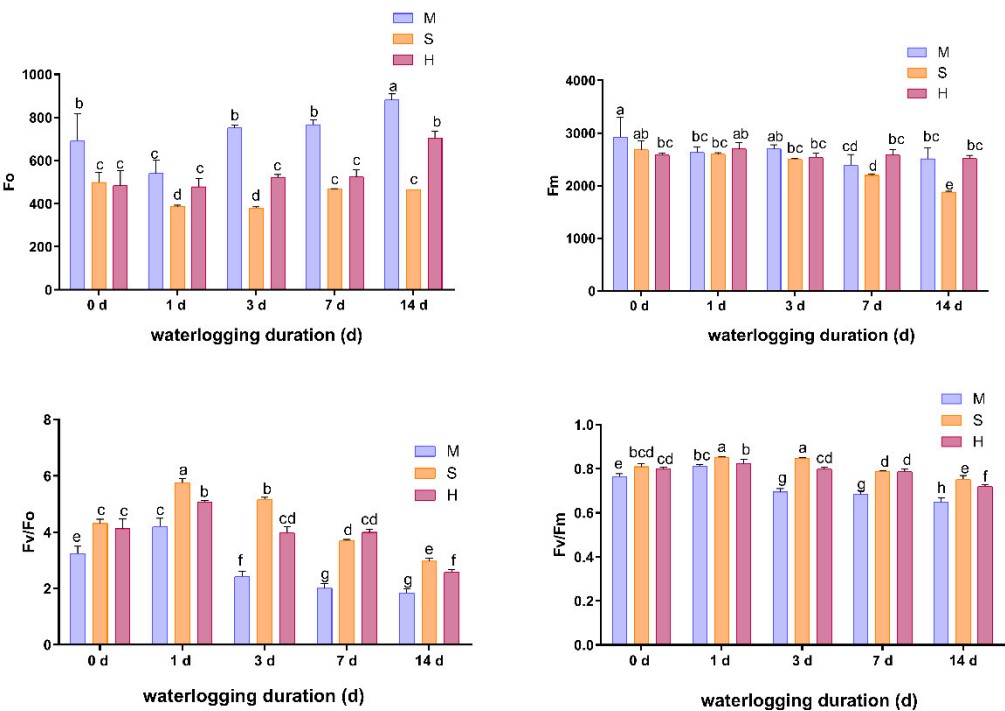

**Figure 4.** The chlorophyll indexes of different peach rootstocks under waterlogging stress. Abbreviations: M: *Prunus persica*, 'Maotao'; S: *Prunus davidiana*, 'Shantao'; H: 'Hossu'. The data are the means ± SDs. Within columns, means with a different letter indicate significant differences (*p* < 0.05) using Duncan's test; bars on the columns represent the standard errors (SD) (n = 6).

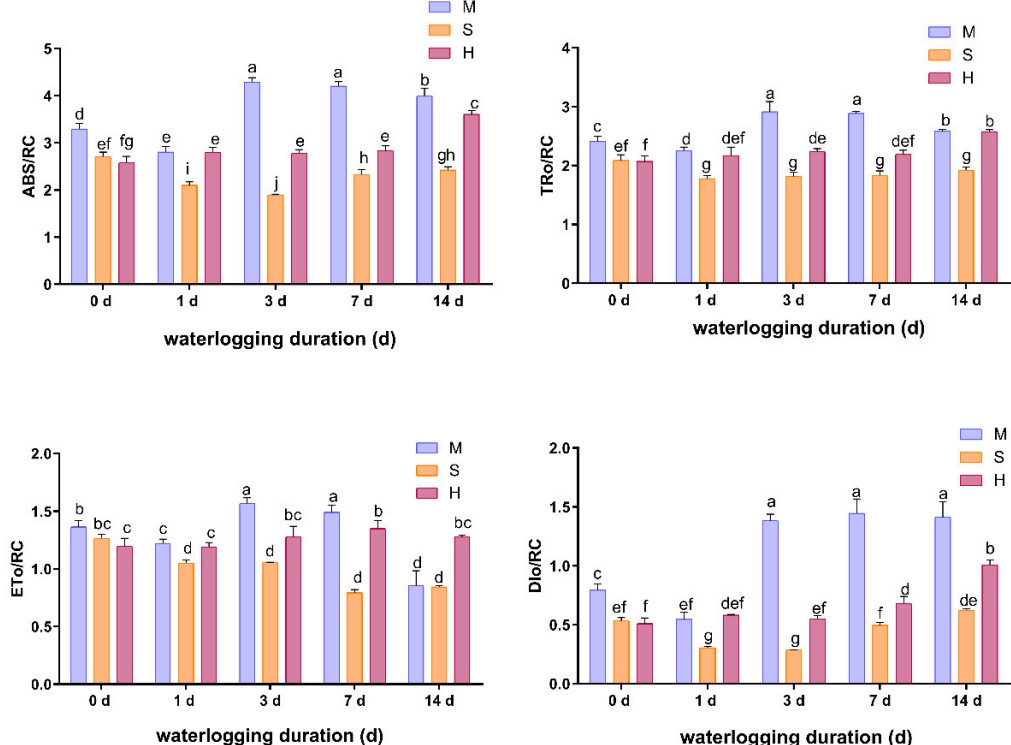

**Figure 5.** Absorption, transformation and usage of light energy in leaf light reaction center of different peach rootstock seedlings under waterlogging stress. Abbreviations: M: Prunus persica, 'Maotao'; S: Prunus davidiana, 'Shantao'; H: 'Hossu'. The data are the means ± SDs. Within columns, means with a different letter indicate significant differences ($p < 0.05$) using Duncan's test; bars on the columns represent the standard errors (SD) (n = 6).

### 3.6. Effects of Waterlogging on Antioxidant and Osmoregulation Responses

As shown in Figure 6, H demonstrated a significant reduction in SOD and CAT activities for 56.1% and 59.9%, respectively, as well as a moderate reduction in POD activity for 17.2% at the initial stage of WT (WT 1 d). The decreases of these enzymes in S were 43.1%, 33.3%, and 35.1%, respectively, while in H they were 5%, 17.1%, and 34.6%. These findings suggested that the capacity of S and H to produce ROS and the capacity of M to scavenge ROS were both severely restricted at the beginning of WT, which may have been the crucial factor for the early over-accumulation of malondialdehyde (MDA) in M.

With the prolongation of waterlogging exposure, the SOD activity of M had become significantly higher in comparison to the non-stressed condition since the third day of WT, while its POD activity increased slightly but remained lower than that of the non-stressed control until the end of WT (WT-14 d).

The SOD activity of S and H both exhibited a similar trend during 1–7 d of WT, increasing before declining. However, during the late stage of WT (WT 7–14 d), the SOD activity of S and H varied, the SOD activity of S remained stable while that of H significantly rebounded, demonstrating that the superoxide free radicals scavenging ability of H was significantly enhanced.

The POD activity of S initially showed a temporary decrease before steadily increasing, whereas that of H decreased gently and constantly. It could be concluded that the coordinated variation of SOD and CAT in H plays a crucial role in ROS scavenging, while SOD and POD served the same purpose in M and S.

Until the 14th day of waterlogging, the leaf proline content of M remained relatively low, but that of S displayed a similar trend with the significant increase that emerged earlier (WT 7 d). The pre-waterlogged leaf proline content of H was higher than that of M and S. Under waterlogging stress, it first declined and then clearly rebounded, and until the

end of WT (WT-14 d), proline content of H was still much lower than that of M and S. This might mean that proline plays a crucial role in preserving the stability of cell structure, particularly in the initial stage of waterlogging.

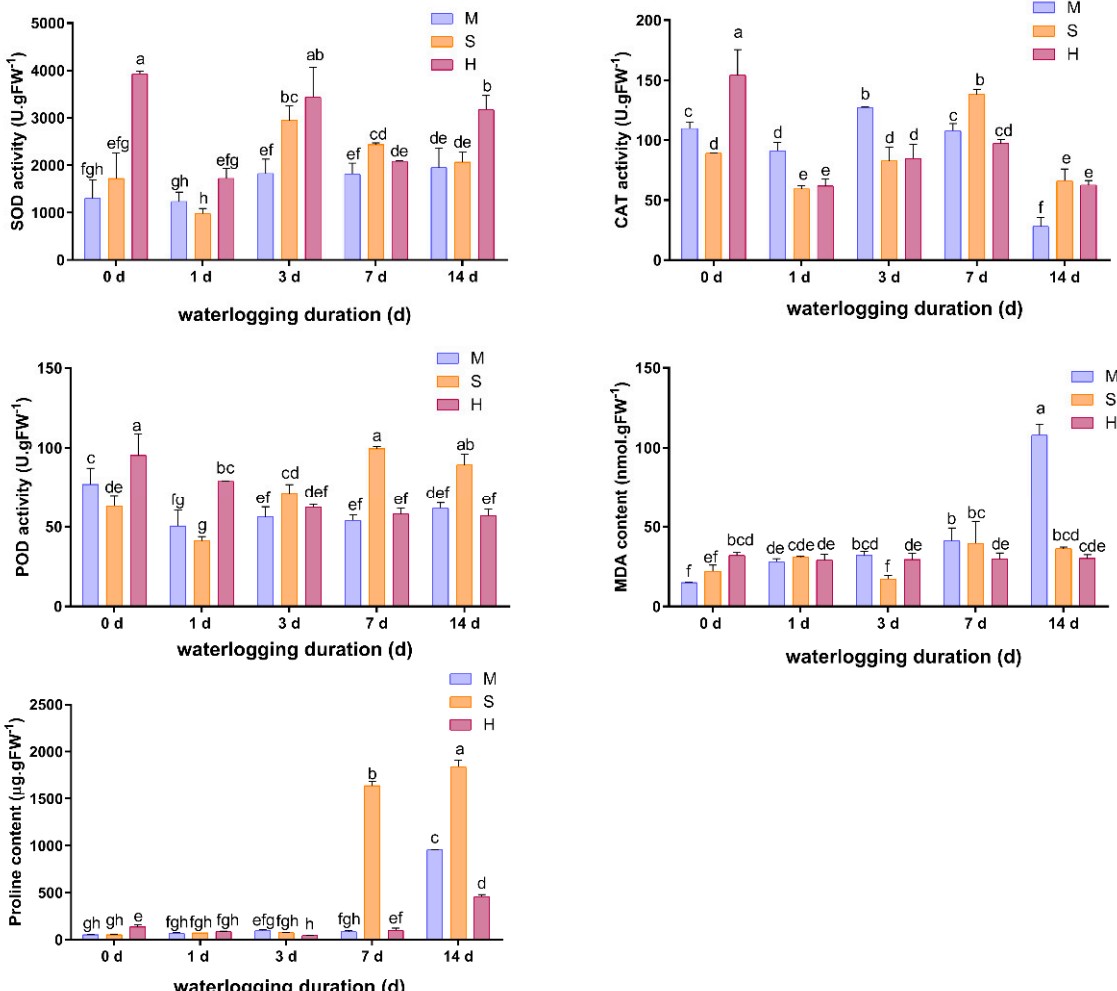

**Figure 6.** The comparison of the activities of antioxidant enzymes and content of MDA and proline of different peach rootstock seedlings under waterlogging stress. Abbreviations: M: Prunus persica, 'Maotao'; S: Prunus davidiana, 'Shantao'; H: 'Hossu'. The data are the means ± SDs. Within columns, means with a different letter indicate significant differences ($p < 0.05$) using Duncan's test; bars on the columns represent the standard errors (SD) (n = 6).

### 3.7. Observation on the Anatomical Structure of Leaves and Fine Roots

3.7.1. Morphological Observations

Under the non-stressed condition, it could be observed that the anatomical structures of leaves of all three tested rootstocks were similar, including the one-layer epidermis and bi-layered compact palisade parenchyma. The palisade tissue of H became single-layered and loosely arranged and its intracellular space increased significantly with the prolongation of waterlogging exposure. In leaf mid-vein and in mesophyll spongy tissue, air cavities primarily initiate from the confluence of the xylem and phloem layer, which was advantageous for air exchange (Figure 7 H-7 and H-14). In M, the palisade parenchyma cell elongated and the toluidine blue (TB) staining diminished with the prolongation of waterlogging exposure. Lower epidermis cells eminence could be observed (Figure 7 M-7). The aggregation and dissipation of starch granules in the phloem parenchyma cells of mid-vein, particularly during the late stage of WT (WT 7–14 d), was the most noticeable change

in the leaf of S under waterlogging treatment. However, no typical air cavity structure could be found in the leaf of S within the overall 14 d of waterlogging.

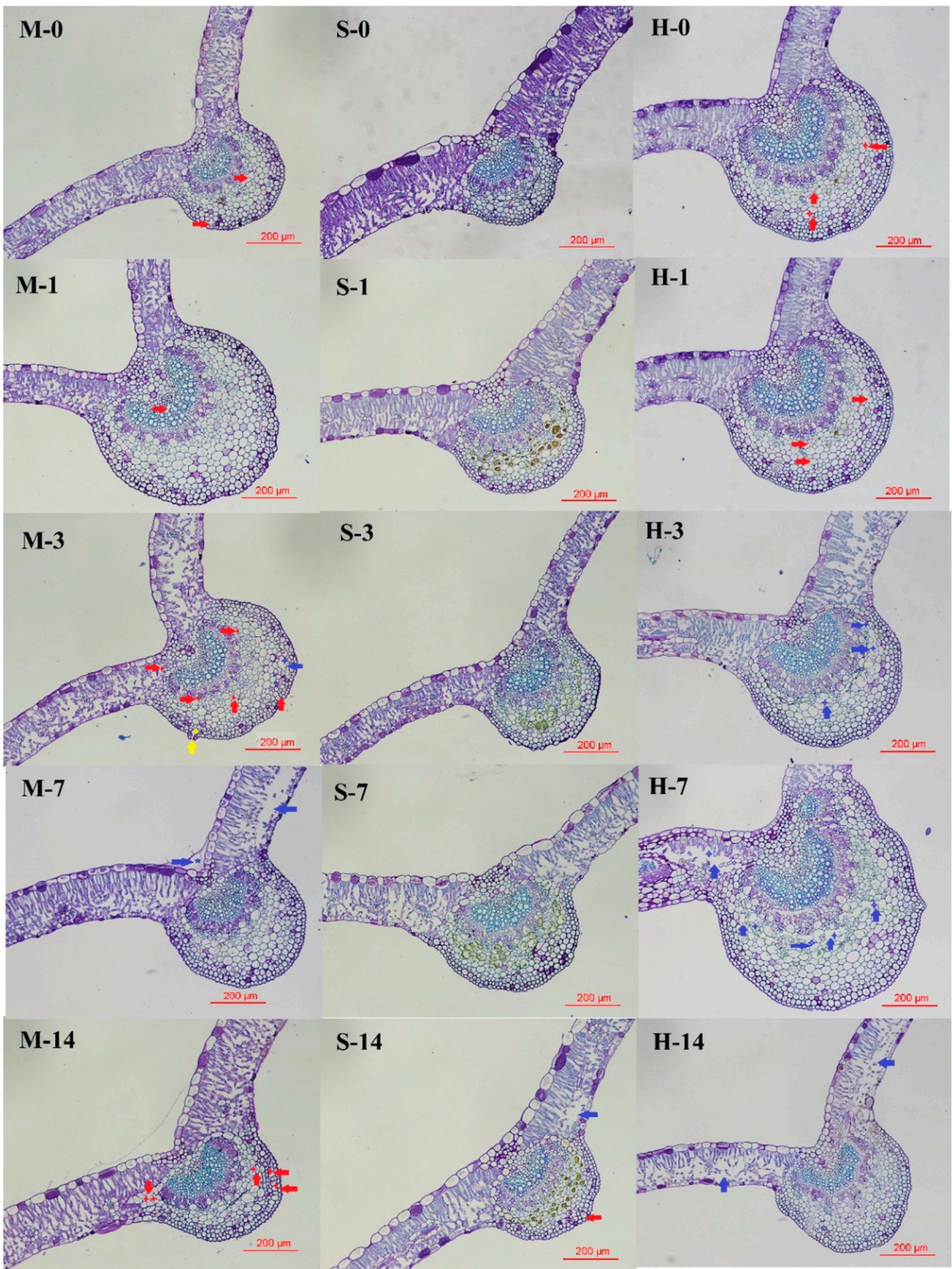

**Figure 7.** Comparison of anatomical structure changes of leaves of different peach rootstocks under different time duration of waterlogging (yellow asterisk/arrow: lenticular structure; blue asterisk/arrow: air cavities; red asterisk/arrow: adjacent cells fused with each other; Bar = 200 µm). Abbreviations: M: Prunus persica, 'Maotao'; S: Prunus davidiana, 'Shantao'; H: 'Hossu'. Leaf samples were collected on un-stressed condition (M-0, S-0, H-0), 1st day (M-1, S-1, H-1), 3rd days (M-3, S-3, H-3), 7th days (M-7, S-7, H-7), and 14th day (M-14, S-14, H-14) of WT.

### 3.7.2. Parameters of Leaf Anatomical Characteristics

Waterlogging stress increased the thickness of the upper epidermis of M by 34.1%, while that of S decreased by 29.9% (Table 3). The upper epidermis thickness of H, in contrast, performed a brief increase before decreasing to its pre-waterlogged level. As opposed to the non-stressed conditions, the thickness of the lower epidermis of M and S exhibited no significant difference, whereas H's performed a transient increase of 7.0% initially and then decreased by 25.6% throughout the 14 d of waterlogging.

**Table 3.** Comparison of leaf anatomical characteristics of different peach rootstock seedlings under waterlogging stress.

| Cultivar | Waterlogged Duration (d) | Upper Epidermis Thickness (μm) | Lower Epidermis Thickness (μm) | Leaf Thickness (μm) | Palisade Cell Length/Width Ratio | CTR% | SR% | P/S Ratio |
|---|---|---|---|---|---|---|---|---|
| M | WT-0 d | 20.39 ± 3.53 b | 10.01 ± 0.84 a | 134.02 ± 2.20 c | 7.23 | 0.34 | 0.37 | 0.92 |
|  | WT-3 d | 21.47 ± 3.76 b | 10.32 ± 0.90 a | 147.76 ± 2.03 b | 6.60 | 0.33 | 0.43 | 0.75 |
|  | WT-14 d | 27.35 ± 3.20 a | 10.97 ± 2.88 a | 177.18 ± 2.72 a | 12.96 | 0.40 | 0.39 | 1.01 |
| S | WT-0 d | 34.03 ± 6.92 a | 12.93 ± 1.30 ab | 185.60 ± 3.05 a | 12.42 | 0.44 | 0.35 | 1.24 |
|  | WT-3 d | 29.88 ± 5.65 ab | 16.33 ± 3.15 a | 172.60 ± 2.69 b | 17.10 | 0.41 | 0.37 | 1.12 |
|  | WT-14 d | 23.87 ± 2.32 c | 13.15 ± 3.29 ab | 118.38 ± 2.47 c | 9.80 | 0.34 | 0.37 | 0.93 |
| H | WT-0 d | 18.86 ± 2.01 b | 12.72 ± 1.34 a | 170.91 ± 2.69 b | 16.77 | 0.39 | 0.36 | 1.09 |
|  | WT-3 d | 21.86 ± 4.33 a | 12.66 ± 2.25 a | 182.92 ± 0.49 a | 10.39 | 0.31 | 0.36 | 0.87 |
|  | WT-14 d | 18.29 ± 1.65 b | 9.46 ± 0.44 b | 141.95 ± 3.51 c | 13.38 | 0.38 | 0.41 | 0.93 |

Abbreviations: M: Prunus persica, 'Maotao'; S: Prunus davidiana, 'Shantao'; H: 'Hossu'. The data are the means ± SDs. Different letters indicate significant differences ($p < 0.05$) using Duncan's test (n = 6).

Additionally, after 14 d of exposure to waterlogging, the leaf thickness of M increased constantly, and its palisade parenchyma cells became longer. Its P/S ratio and CTR% increased by 9.8% and 17.6%. In contrast, the decreases in leaf thickness, P/S ratio, and CTR% of S within 14 d were 36.2%, 25%, and 22.7%, respectively, with its palisade parenchyma cells becoming shorter and thicker. The leaf thickness and P/S ratio of H decreased by 16.9% and 14.6%, respectively, and its CTR% decreased by 20.5% on the third day of WT before returning to a non-stressed level.

### 3.7.3. Observation of the Anatomical Structure of Fine Roots

As shown in Figure 8, it could be observed that the cross-sections of fine roots of M and H shared similar one-layer exterior epidermal cells, while that of S was formed by multi-layer epidermal cells (Figure 8 M-0, S-0, H-0).

The fine roots of M atrophied significantly with the prolongation of waterlogging exposure, whereas those of S and H swelled, which is probably due to the swelling of aerenchyma development in parenchyma cells. During the late stage of waterlogging (WT 7–14 d), the cortical parenchyma cells of M were significantly distorted, whereas those of S expanded and exfoliated (Figure 8 M-14, S-14). This demonstrated that, in comparison to M and S, the appropriate location sites of air cavities in H have a less negative impact on the integrity and stability of its cell structure, thereby increasing the oxygen permeability in its fine roots to a greater extent.

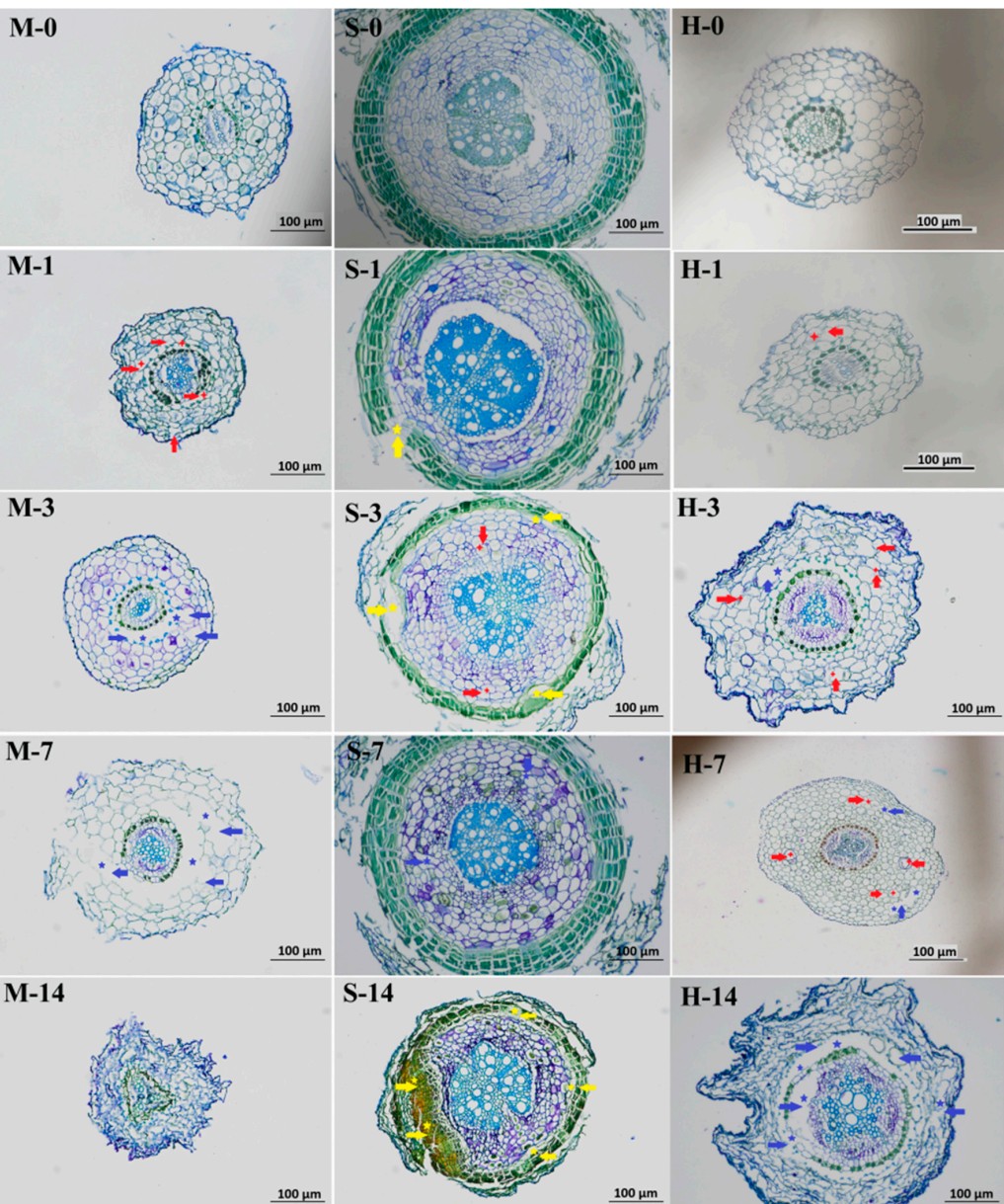

**Figure 8.** Comparison of anatomical structure changes of fine roots of different peach rootstocks under different time duration of waterlogging (yellow asterisk/arrow: lenticular structure; blue asterisk/arrow: air cavities; red asterisk/arrow: adjacent cells fused with each other; Bar = 200 μm). Abbreviations: M: Prunus persica, 'Maotao'; S: Prunus davidiana, 'Shantao'; H: 'Hossu'. Fine root samples were collected on non-stressed condition (M-0, S-0, H-0), 1st day (M-1, S-1, H-1), 3rd day (M-3, S-3, H-3), 7th day (M-7, S-7, H-7), and 14th day (M-14, S-14, H-14) of WT.

### 3.7.4. Parameters of Fine Roots Anatomical Characters

Table 4 demonstrates that during the 14 days of waterlogging treatment, the peripheral epidermis and phloem layer of fine roots of S significantly swelled, increasing in thickness by 80.4% and 60.9%, respectively. The phloem layer thickness of H initially decreased and then increased by 26.6% compared to that under the non-stressed conditions. The phloem thickness of M briefly decreased at first and then returned to the non-stressed level at the end of waterlogging. In contrast, the diameter of the root columns of M and S expanded by 35.1% and 78.0%, respectively, while that of H increased transiently before decreasing to the non-stressed level. Due to the tighter cell arrangement in the root column than

that of the phloem, it could be concluded that waterlogging stress may reduce the oxygen permeability of roots of M and S, whereas that of H was enhanced.

**Table 4.** Anatomical characteristics of the fine roots of different peach rootstocks under waterlogging stress.

| Cultivar | Waterlogged Duration (d) | Thickness of Peripheral Epidermis (μm) | Thickness of Phloem (μm) | The Diameter of Root Column (μm) |
|---|---|---|---|---|
| M | WT-0 d | - | 101.41 ± 1.95 a | 55.35 ± 10.62 c |
|   | WT-3 d | - | 104.74 ± 15.38 a | 104.67 ± 15.25 a |
|   | WT-14 d | - | 105.61 ± 3.32 a | 74.77 ± 1.53 c |
| S | WT-0 d | 31.57 ± 8.58 bc | 69.86 ± 5.88 c | 156.52 ± 15.68 b |
|   | WT-3 d | 39.15 ± 2.98 b | 91.33 ± 3.68 b | 175.87 ± 7.60 b |
|   | WT-14 d | 56.96 ± 4.05 a | 112.38 ± 30.48 a | 278.54 ± 5.24 a |
| H | WT-0 d | - | 177.82 ± 14.18 b | 124.52 ± 6.61 b |
|   | WT-3 d | - | 82.70 ± 4.43 c | 205.14 ± 28.05 a |
|   | WT-14 d | - | 225.09 ± 15.21 a | 127.49 ± 15.17 b |

Abbreviation: M: Prunus persica, 'Maotao'; S: Prunus davidiana, 'Shantao'; H: 'Hossu'. The data are the means ± SDs. A different letter indicates significant differences ($p < 0.05$) using Duncan's test (n = 6).

### 3.8. SEM Observation of Stems and Roots

The anatomical structures of stems and primary lateral roots were examined using SEM under non-stressed (WT-0 d) and waterlogged conditions (WT-7 d). In stems, the cortical parenchyma cells of S swelled more significantly than M and H, and larger salt crystals generation and greater starch granule loss could be observed among the parenchyma cell layers of both M and S after 7 days of WT (Figures 9 and 10). Under the same level of waterlogging stress, larger quantities of starch granules could still be observed in the cortical parenchyma cells of H, which maintained relatively constant peripheral cortex structure. Secondly, waterlogging stress might result in the loss of starch granules in both stems and roots; after 7 days of WT, S showed much more severe starch granule loss in both pith and xylem ray cells than those of M and H. Since starch granule hydrolysis may be involved in energy metabolism under hypoxic/anaerobic conditions, this means that S had significantly lower levels of the energy metabolism than those of M and H under waterlogging. Osmotic regulation may also be influenced by the dissolution and precipitation of salt crystal particles. The increased size of the salt crystal particles of H may therefore contribute to its stronger osmotic regulatory capability.

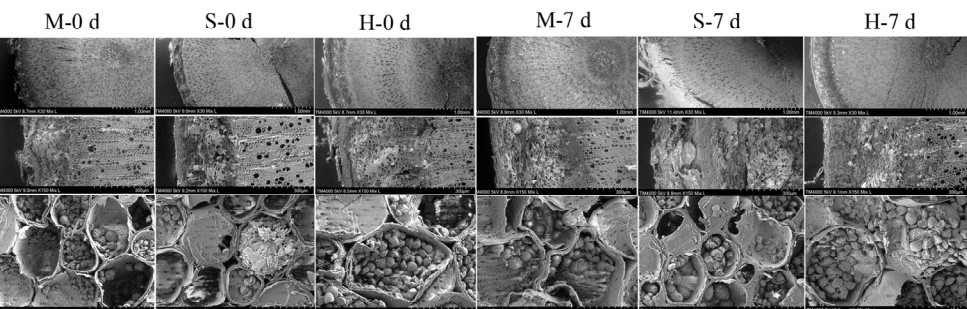

**Panoramic view of stem (×30)**

**Peripheral cortex (×150)**

**Starch granule in pith cells (×1.2k)**

**Figure 9.** SEM observation and comparison of anatomical structure changes of stem (cross-section) of different peach rootstocks, the biggest differences between the two treatments (un-stressed control and 7 d of waterlogging) are the morphology of the peripheral cortex layer, and the amount of starch granules in the pith cells. Abbreviations: M: Prunus persica, 'Maotao'; S: Prunus davidiana, 'Shantao'; H: 'Hossu'. Stem samples were collected under non-stressed condition (M-0, S-0, H-0) and on the 7th day (M-7, S-7, H-7) of WT. "×30", "×150", and "×1.2 k" refer to different magnifications of SEM when observing different parts of samples.

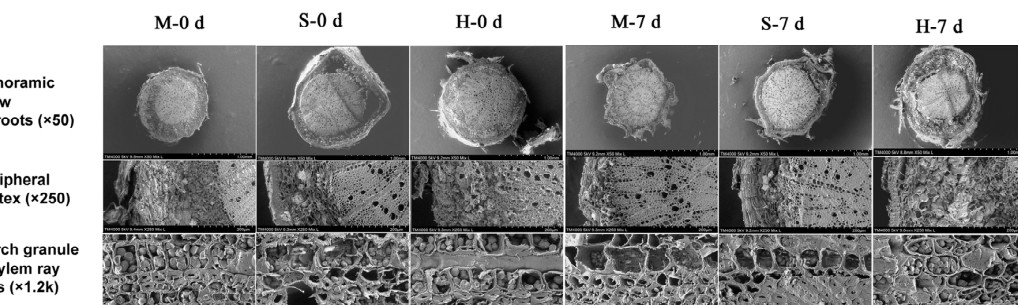

**Figure 10.** SEM observation and comparison of anatomical structure changes of roots (cross-section) of different peach rootstocks; the biggest differences between the two treatments (un-stressed control and 7 d of waterlogging) are the morphology of the peripheral cortex layer and the amount of starch granules in the xylem ray cells and pith cells. Abbreviations: M: Prunus persica, 'Maotao'; S: Prunus davidiana, 'Shantao'; H: 'Hossu'. Root samples were collected under non-stressed condition (M-0, S-0, H-0) and on the 7th day (M-7, S-7, H-7) of WT. "×50", "×250", and "×1.2 k" refer to different magnifications of SEM when observing different parts of samples.

### 3.9. TEM Observation of Leaves and Roots

As shown in Figure 11, the ultra-structure of the chloroplasts in leaves was integrated and it clung to the cell walls, appearing as long, narrow bands with spindle or oval shapes and the large size starch granule inside under non-stressed condition (Figure 11 M-0, S-0, H-0). Until the 7th day of waterlogging treatment, some chloroplasts and nucleolus of M degraded (Figure 11 M-7-1) and some of them were expanded, not adhering to cell walls but moving to the central zone of the cell (Figure 11 M-7-2). In leaves of S, its chloroplasts also moved to the central zone of the cell and became large and round with the nucleolus degraded (Figure 11 S-7-1). The chloroplasts of H, however, only slightly expanded (Figure 11 H-7-2). Meanwhile, higher rate of intracellular mycorrhizal fungal infection (Figure 11, red and yellow asterisks) and intracellular hyphal connection (Figure 11, blue arrows) could be observed in the leaves of both M and H with the prolongation of waterlogging exposure, but not in S.

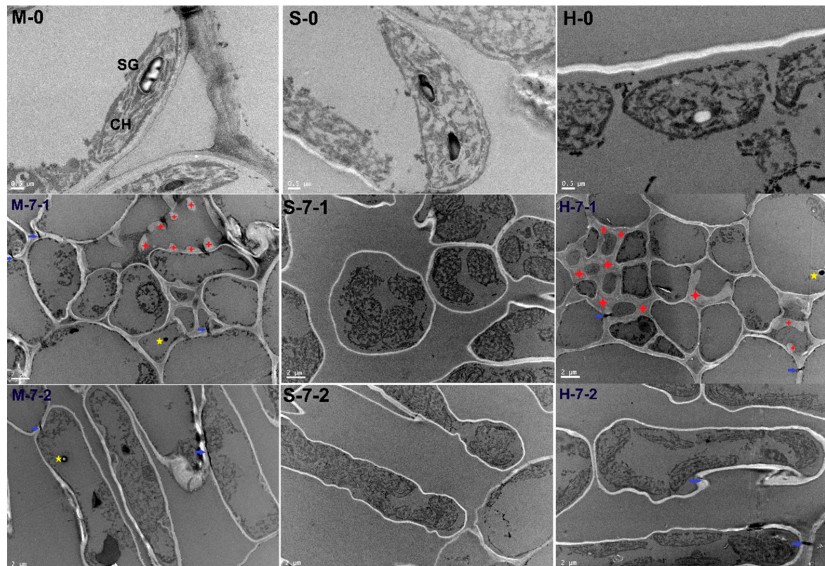

**Figure 11.** Comparison of cell ultra-structure of leaves of different peach rootstocks exposed to 7 d of waterlogging stress under TEM (SG: starch granule; CH; chloroplast; red and yellow asterisk: mycorrhizal fungi infection; blue arrow: mycelium; bar = 0.5 μm (M-0, S-0, H-0); bar = 2 μm (M-7-1 and M-7-2, S-7-1 and S-7-2, H-7-1 and H-7-2)). Abbreviations: M: Prunus persica, 'Maotao'; S: Prunus davidiana, 'Shantao'; H: 'Hossu'. Leaf samples were collected under un-stressed condition (M-0, S-0, H-0) and on the 7th day (M-7-1 and M-7-2, S-7-1 and S-7-2, H-7-1 and H-7-2) of WT.

Under non-stressed condition, fine root cells of all three tested rootstocks displayed mycorrhizal fungus infection (Figure 12 M-0, S-0, H-0, red asterisk) and intercellular hyphal junction (Figure 12 M-0, S-0, H-0, blue arrow). The rate of intracellular mycorrhizal infection (Figure 12 M-7, S-7, H-7, red asterisk) and the hyphal connection between adjacent cells (Figure 12 M-7, S-7, H-7, blue arrow) in the fine roots of H rose the greatest until the 7th day of waterlogging treatment, whereas that of S altered the least.

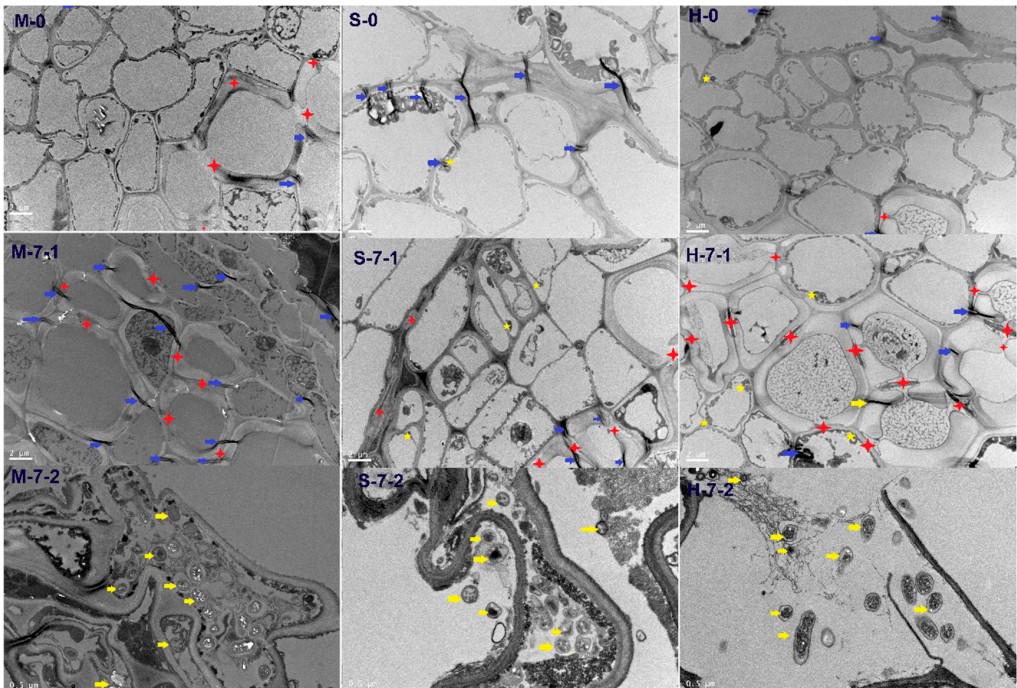

**Figure 12.** Comparison of ultra-structure of fine roots of different peach rootstocks exposed to 7 d of waterlogging stress under TEM (red and yellow asterisk: mycorrhizal fungi infection; blue arrow: mycelium; yellow arrow: bacteria, actinomycetes or fungal spores; bar = 0.5 μm (M-0, S-0, H-0); bar = 2 μm (M-7-1 and M-7-2, S-7-1 and S-7-2, H-7-1 and H-7-2)). Abbreviations: M: Prunus persica, 'Maotao'; S: Prunus davidiana, 'Shantao'; H: 'Hossu'. Leaf samples were collected under un-stressed condition (M-0, S-0, H-0) and on the 7th day (M-7-1 and M-7-2, S-7-1 and S-7-2, H-7-1 and H-7-2) of WT.

### 3.10. Evaluation of Waterlogging Tolerance of Different Peach Rootstocks

According to the results of a principal component analysis, plant height (PH, contribution rate of 29.74%), stem diameter (SD, contribution rate of 26.587%), upper (contribution rate of 10.01%) and lower part (contribution rate of 7.683%) of leaf SPAD values, as well as three photosynthetic parameters ($C_i$, 4.871%, $P_n$, 4.379%, and $g_s$, 4%) are best represented (totally 87.271%) cumulative contribution (Table 5). The heatmap also confirmed that these seven main components correlated significantly (Pearson correlation coefficients) (Figure 13).

Additionally, a membership function analysis was carried out to assess the variance in waterlogging tolerance of the three rootstocks. According to Table 6, H showed much stronger tolerance over the course of the 14 days of waterlogging therapy compared to M and S. However, its overall waterlogging tolerance (WT-14 d) showed no difference from S, both of which were significantly lower than that of H. Furthermore, M's comprehensive tolerance within 3 days of waterlogging (M-3 d ranked 3rd) was obviously better than that of S (S-3 d ranked 6th), which might indicate that M can better tolerate short-term waterlogging stress (within 3 days).

**Table 5.** Evaluation of waterlogging tolerance of different peach rootstocks based on a principal component analysis.

| Principal Component | Parameters | Initial Eigenvalue | Total Variance of Interpretation | |
|---|---|---|---|---|
| | | | Contribution Rate % | Cumulative Contribution Rate % |
| 1 | PH (Plant Height) | 8.03 | 29.74 | 29.74 |
| 2 | SD (Stem Diameter) | 7.178 | 26.587 | 56.327 |
| 3 | SPAD upper | 2.703 | 10.01 | 66.337 |
| 4 | SPAD lower | 2.074 | 7.683 | 74.021 |
| 5 | Ci | 1.315 | 4.871 | 78.892 |
| 6 | Pn | 1.182 | 4.379 | 83.272 |
| 7 | gs | 1.08 | 4 | 87.271 |
| 8 | E | 0.983 | 3.639 | 90.911 |
| 9 | WUE | 0.549 | 2.032 | 92.942 |
| 10 | Ls | 0.424 | 1.57 | 94.513 |
| 11 | Fo | 0.317 | 1.175 | 95.688 |
| 12 | Fm | 0.239 | 0.886 | 96.574 |
| 13 | Fv/Fm | 0.229 | 0.848 | 97.421 |
| 14 | Fv/Fo | 0.186 | 0.69 | 98.111 |
| 15 | ABS/RC | 0.126 | 0.467 | 98.578 |
| 16 | TRo/RC | 0.112 | 0.415 | 98.993 |
| 17 | ETo/RC | 0.084 | 0.31 | 99.302 |
| 18 | DIo/RC | 0.047 | 0.175 | 99.477 |
| 19 | PIabs | 0.041 | 0.152 | 99.629 |
| 20 | PItotal | 0.031 | 0.114 | 99.743 |
| 21 | SOD | 0.029 | 0.106 | 99.849 |
| 22 | CAT | 0.015 | 0.054 | 99.904 |
| 23 | POD | 0.011 | 0.041 | 99.945 |
| 24 | MDA | 0.006 | 0.022 | 99.968 |
| 25 | Proline | 0.005 | 0.018 | 99.985 |
| 26 | TTC | 0.004 | 0.014 | 100 |
| 27 | Root respiration rate | $7.62 \times 10^{-5}$ | 0 | 100 |

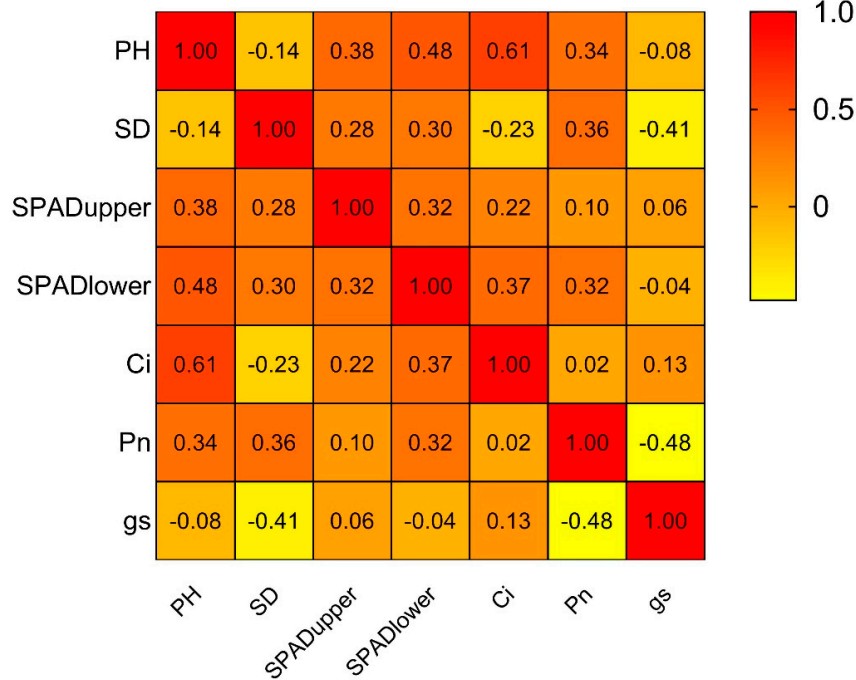

**Figure 13.** The correlation analysis heatmaps of top seven principal components (Pearson correlation coefficients).

**Table 6.** Evaluation of waterlogging tolerance of different peach rootstocks based on membership function analysis.

| Parameters | M-0 d | M-1 d | M-3 d | M-7 d | M-14 d | S-0 d | S-1 d | S-3 d | S-7 d | S-14 d | H-0 d | H-1 d | H-3 d | H-7 d | H-14 d |
|---|---|---|---|---|---|---|---|---|---|---|---|---|---|---|---|
| Plant Height | 0.21 | 0.26 | 0.31 | 0.25 | 0.35 | 0.83 | 0.89 | 0.92 | 0.93 | 1.00 | 0.02 | 0.00 | 0.05 | 0.12 | 0.20 |
| Stem Diameter | 0.10 | 0.26 | 0.11 | 0.00 | 0.17 | 0.69 | 0.85 | 0.70 | 0.34 | 0.47 | 0.65 | 1.00 | 0.66 | 0.81 | 0.86 |
| SPAD upper | 0.45 | 0.54 | 0.50 | 0.50 | 0.00 | 0.70 | 0.82 | 1.00 | 0.95 | 0.68 | 0.68 | 0.69 | 0.79 | 0.86 | 0.79 |
| SPAD lower | 0.53 | 0.12 | 0.34 | 0.13 | 0.00 | 0.21 | 0.21 | 0.40 | 0.42 | 0.50 | 0.26 | 0.92 | 0.94 | 1.00 | 0.59 |
| Ci | 0.35 | 0.10 | 0.00 | 0.32 | 0.31 | 0.12 | 0.73 | 0.32 | 1.00 | 0.49 | 0.38 | 0.26 | 0.31 | 0.24 | 0.67 |
| Pn | 0.75 | 0.53 | 0.45 | 0.15 | 0.09 | 0.68 | 0.18 | 0.28 | 0.00 | 0.17 | 1.00 | 0.67 | 0.74 | 0.16 | 0.12 |
| gs | 0.67 | 0.29 | 0.17 | 0.00 | 0.03 | 0.61 | 0.03 | 0.05 | 0.03 | 0.05 | 1.00 | 0.39 | 0.30 | 0.01 | 0.00 |
| E | 0.88 | 0.52 | 0.40 | 0.01 | 0.04 | 0.21 | 0.09 | 0.18 | 0.05 | 0.10 | 1.00 | 0.58 | 0.88 | 0.04 | 0.00 |
| WUE | 0.82 | 0.94 | 0.94 | 0.71 | 0.46 | 1.00 | 0.61 | 0.57 | 0.00 | 0.70 | 0.92 | 0.87 | 0.85 | 0.70 | 0.72 |
| Ls | 0.65 | 0.90 | 1.00 | 0.68 | 0.69 | 0.88 | 0.27 | 0.68 | 0.00 | 0.51 | 0.62 | 0.74 | 0.69 | 0.75 | 0.33 |
| Fo | 0.62 | 0.32 | 0.74 | 0.77 | 1.00 | 0.24 | 0.01 | 0.00 | 0.17 | 0.17 | 0.21 | 0.20 | 0.28 | 0.29 | 0.65 |
| Fm | 1.00 | 0.72 | 0.79 | 0.49 | 0.60 | 0.77 | 0.69 | 0.60 | 0.30 | 0.00 | 0.68 | 0.79 | 0.63 | 0.67 | 0.61 |
| Fv/Fm | 0.57 | 0.80 | 0.23 | 0.17 | 0.00 | 0.79 | 1.00 | 0.98 | 0.69 | 0.51 | 0.74 | 0.86 | 0.74 | 0.68 | 0.35 |
| Fv/Fo | 0.36 | 0.60 | 0.15 | 0.04 | 0.00 | 0.63 | 1.00 | 0.85 | 0.48 | 0.29 | 0.58 | 0.83 | 0.55 | 0.55 | 0.19 |
| ABS/RC | 0.58 | 0.38 | 1.00 | 0.96 | 0.88 | 0.34 | 0.09 | 0.00 | 0.18 | 0.22 | 0.29 | 0.38 | 0.37 | 0.39 | 0.71 |
| TRo/RC | 0.55 | 0.42 | 1.00 | 0.97 | 0.71 | 0.27 | 0.00 | 0.03 | 0.05 | 0.13 | 0.26 | 0.35 | 0.40 | 0.36 | 0.70 |
| ETo/RC | 0.74 | 0.55 | 1.00 | 0.90 | 0.08 | 0.61 | 0.33 | 0.34 | 0.00 | 0.07 | 0.52 | 0.52 | 0.63 | 0.72 | 0.63 |
| DIo/RC | 0.44 | 0.22 | 0.95 | 1.00 | 0.97 | 0.21 | 0.02 | 0.00 | 0.18 | 0.29 | 0.19 | 0.25 | 0.23 | 0.34 | 0.62 |
| PIabs | 0.18 | 0.29 | 0.12 | 0.05 | 0.00 | 0.38 | 0.79 | 1.00 | 0.19 | 0.15 | 0.35 | 0.32 | 0.37 | 0.32 | 0.10 |
| PItotal | 0.41 | 0.29 | 0.43 | 0.13 | 0.00 | 0.18 | 0.53 | 0.61 | 0.04 | 0.04 | 1.00 | 0.63 | 0.56 | 0.34 | 0.16 |
| SOD | 0.11 | 0.09 | 0.29 | 0.28 | 0.33 | 0.25 | 0.00 | 0.67 | 0.49 | 0.37 | 1.00 | 0.25 | 0.83 | 0.38 | 0.74 |
| CAT | 0.65 | 0.50 | 0.79 | 0.63 | 0.00 | 0.49 | 0.25 | 0.43 | 0.87 | 0.30 | 1.00 | 0.27 | 0.45 | 0.55 | 0.27 |
| POD | 0.62 | 0.16 | 0.26 | 0.22 | 0.35 | 0.38 | 0.00 | 0.52 | 1.00 | 0.83 | 0.93 | 0.65 | 0.36 | 0.29 | 0.27 |
| MDA | 0.00 | 0.14 | 0.18 | 0.28 | 1.00 | 0.08 | 0.17 | 0.02 | 0.26 | 0.23 | 0.18 | 0.15 | 0.16 | 0.16 | 0.16 |
| Proline | 0.00 | 0.01 | 0.03 | 0.02 | 0.51 | 0.00 | 0.01 | 0.02 | 0.89 | 1.00 | 0.05 | 0.02 | 0.00 | 0.03 | 0.23 |
| TTC | 0.10 | 0.21 | 1.00 | 0.56 | 0.06 | 0.07 | 0.14 | 0.55 | 0.14 | 0.00 | 0.29 | 0.60 | 0.86 | 0.32 | 0.27 |
| Root respiration | 0.24 | 0.27 | 1.00 | 0.45 | 0.19 | 0.46 | 0.00 | 0.55 | 0.10 | 0.02 | 0.52 | 0.09 | 0.62 | 0.39 | 0.21 |
| Average value | 0.47 | 0.39 | 0.52 | 0.40 | 0.33 | 0.45 | 0.36 | 0.45 | 0.36 | 0.34 | 0.57 | 0.49 | 0.53 | 0.43 | 0.41 |
| Ranking * | 5 | 10 | 3 | 9 | 13 | 6 | 11 | 6 | 11 | 12 | 1 | 4 | 2 | 7 | 8 |

* Comprehensive ranking of waterlogging tolerance of different peach rootstocks under different time duration of waterlogging treatments.

## 4. Discussion

The development responses of peach rootstocks to waterlogging were influenced by both genetic and environmental factors [15,36,37]. For example, Iacona et al. [38] found that the adoption of a grafted combination with a cloned variety of a novel flood-tolerant stone-fruit rootstocks might increase the survival rate of cultivated peaches during anoxia conditions caused by waterlogging. There are currently limited resources and relevant approaches for identifying or breeding peach rootstocks that can withstand waterlogging. Therefore, in this work, photochemical, physiological, and anatomical response mechanisms of different peach rootstocks were comprehensively investigated to compare their waterlogging tolerance mechanisms.

### 4.1. Morphological Responses

The results of the present study showed that after 14 days of waterlogging, plant mortality of S and M was 91.6% and 58.3%, respectively (data not shown), but all tested seedlings of H survived. This suggests that waterlogging can decrease the survival rate of peach rootstocks. The slowing of shoot growth, leaf yellowing, dehydration, and defoliation are the most prominent visible effects of short-term waterlogging; similar results were reported in sorghum by Zhang et al. [27] and in maize by Zhu et al. [29], which might indicate that the inhibitory effect of waterlogging on leaf photosynthetic performance is primarily due to the destruction of photosynthetic pigments dominated by chlorophyll.

### 4.2. Photosynthetic Physiological Responses

Stomatal closure was widely considered as one of the fastest responses of different plants to waterlogging stress. This response usually causes a correlated decrease in the Pn and E, which may mean that the photosynthesis of plants has been impaired since the very beginning of waterlogging. Results of the present study presented that, under waterlogging, the values of WUE, Pn, and gs of three rootstocks generally decreased, whereas the Ci value showed divergent tendencies. The findings of this study were consistent with previous research in that all three peach rootstocks decreased their Pn, gs, and E values [1,39,40]. However, the Ci value change trend differs, indicating that stomatal and non-stomatal factors caused similar intracellular $CO_2$ assimilation processes in M and H during the early and late stages of waterlogging, respectively [1,5,41], whereas $CO_2$ assimilation in S is more affected by non-stomatal factors; especially during the later stage of WT, its photosynthetic metabolism was almost irreversible damaged. This result closely corresponds to the phenomenon of browning, and massive abscission of peach leaves during the final stages of waterlogging treatment.

Chlorophyll degradation, which was considered to be the primary cause of waterlogging-induced leaf yellowing, may reduce light absorption, avoid photo-oxidation [1,42], and lead to photosynthesis system destruction [12,42]. Similar results were obtained on various *Prunus* rootstocks [20,43,44], sorghum [27], and maize [10], indicating that these plants adapt to waterlogging stress by decreasing the leaf Chl a content and the electron transport rate of the photosynthetic electron transport chain. However, the present research has yielded inconsistent conclusions that under the same degree of waterlogging stress, H with better waterlogging tolerance showed relatively higher chlorophyll content (Figure 1) and photosynthetic electron transfer efficiency (Figure 5) as compared to that of M and S. Xiao et al. [4] reported results similar to this study that exogenous $H_2S$ application alleviates the waterlogging injuries on peach seedlings through reducing ROS accumulation in roots and leaves, which is mainly achieved by increasing chlorophyll content and photosynthetic capacity. All these diverse conclusions may mean that peach seedlings and field crops have varied photosynthetic response mechanisms to waterlogging.

### 4.3. Antioxidant Defense Responses

Reactive oxygen species (ROS) are the main toxic byproducts during various metabolic reactions, such as photosynthesis and respiration. Plants have evolved an elaborate system

of enzymatic and non-enzymatic antioxidants that help to scavenge these over-accumulated ROS and reduce oxidative damage under waterlogging stress [10,11,14,45,46]. It was widely assumed that SOD converts $O_2^-$ to $H_2O_2$, followed by CAT and POD, which primarily convert $H_2O_2$ to $H_2O$ and $O_2$ thereby reducing oxidative injuries [47]. For example, Gu et al. [6] confirmed that melatonin-treated Prunus persica plants had higher levels of SOD and POD activity as well as lower levels of MDA and $H_2O_2$ contents. Thus, its waterlogging tolerance enhanced. McGee et al. [44] found that waterlogging increased antioxidant activities, osmolyte concentrations, MDA, and ROS accumulation, while decreasing leaf nutrient content of all six Prunus spp. rootstocks tested. According to the findings of this study, it could be concluded that the antioxidant defense system of peach rootstocks was significantly activated in a very short time of waterlogging, and it could effectively protect the plants from severe oxidative damages, but the ROS level regulatory mechanisms in these three rootstocks under waterlogging stress varied: the crucial point of H's antioxidant defense system was to inhibit the generation of $H_2O_2$, whereas M and S focused on promoting the decomposition of $H_2O_2$ (Figure 6).

### 4.4. Osmoregulation Responses

MDA is the primary, toxic byproduct of membrane lipid peroxidation. Proline is an osmotic substance that, in addition to regulating intracellular osmotic balance, eliminates free radicals and protects cells from oxidative damage by maintaining structural integrity [32,44]. Many previous studies suggested that an increase in MDA content indicates membrane structure destruction and an increase in membrane permeability, whereas an increase in intracellular proline level under waterlogging stress is primarily involved in maintaining cell osmotic pressure and, to a certain extent, protecting the membrane from oxidative injuries [4,48]. Moreover, proline was also thought to be involved in ROS scavenging [49]. In this study, however, the accumulation level in proline content of s in the late stage of waterlogging treatment was significantly higher than that of M and H (Figure 6), but the survival rate and waterlogging performance of S were worse than M and H. This finding differs from the majority of previous research. Most researchers believe that a high proline content indicates good osmotic adjustment and stress resistance. We therefore speculated that in waterlogged S seedlings, this could result in irreversible damage, such as cell disintegration and intracellular ion leakage.

### 4.5. Anatomical and Ultrastructural Responses

The most common anatomical responses to waterlogging include the formation of aerenchyma and hypertrophied lenticels [2,50,51], which are typically initiated from the peripheral cortex layer [52], parenchyma cells of the phloem [2,10], or column tissue [10,46] of roots and stems. The roots, stems, and leaves displayed a variety of changes in anatomical structures to adapt to waterlogged conditions. The generation of aerenchyma is typically thought to be beneficial to improve plants' waterlogging tolerance [50,52]. For example, Peng et al. [2] compared two full-sib poplar clones and found that a larger size of aerenchyma appeared in roots of flood-susceptible genotype and caused more severe root anatomical structural deformation than those of flood-tolerant ones. Furthermore, according to Salah et al. [10], exogenous applied spermidine and brassinosteroid can improve the waterlogging tolerance of maize seedlings by reducing root aerenchyma area and scavenging intracellular ROS. In contrast, the present study showed a quite different result that during 14 days of WT, a greater quantity and size of air cavities generated from cortical parenchyma cells could be observed in roots and mesophyll of H rather than M and S (Figures 7 and 8). This is probably because of the appropriate location of air cavities in root, stem, and leaf tissue of H, which contributes to its better tissue stability and relatively higher survival rate under waterlogging. Similar results were reported by Purnobasuki et al. [21] in tobacco.

Further exploration was carried out by SEM technologies in this study, and the findings revealed that after 7 days of WT, H preserved much more starch granules than that of M and S, with most of the starch granules was in the xylem ray cells and pith cells of roots and

stems (Figures 9 and 10). This is consistent with previous findings reported by Omi and Robin [53] and Gravatt and Kirby [39] that maintaining relatively greater pre-waterlogged root tissue starch granules accumulation is a crucial component enabling flood-tolerant species to survive in waterlogging stress.

Our findings from TEM observations of leaves and roots suggested a possible correlation between mycorrhizal fungi infection and waterlogging tolerance of peach rootstocks. It is traditionally believed that mycorrhizal symbiosis participates in proline metabolism and improves root morphology, which can improve the tolerance of plants to various abiotic stresses [4,31,54,55]. This effect is more pronounced in juvenile seedlings [32]. In comparison, based on the significant increase in proline accumulation (Figure 6) and TEM observation results of leaf samples (Figure 11) on the 7th day of WT, we can draw a similar conclusion that an increase in AMF infection rate in mesophyll palisade cells may be associated with an increase in waterlogging tolerance in H and M to varying degrees, while S with relatively lower waterlogging tolerance showed no obvious changes in AMF infection rate in mesophyll cells or fine roots.

## 5. Conclusions

Consequently, rootstock H's greater increased photosynthetic electron transfer efficiency, better improved oxygen permeability due to earlier air cavity formation in both leaf midvein and fine roots, and higher energy metabolism due to greater starch granule accumulation may all contribute to its higher tolerance to waterlogging as compared to M and S. Correlation analysis and membership function analysis of all principal components revealed that the three rootstocks differed in their waterlogging tolerance. More research is needed, however, to determine the fundamental relationship between mycorrhizal symbiosis and aerenchyma development in roots and leaves under waterlogging stress.

**Author Contributions:** Conceptualization, Z.Y. and C.G.; methodology, J.D.; validation, X.Z., X.L. and Y.H.; formal analysis, F.X.; investigation, H.C.; resources, J.D.; data curation, H.C., M.S., H.Z. and M.Z.; writing—original draft preparation, F.X. and H.C.; writing—review and editing, F.X.; supervision, J.D. and Z.Y.; project administration, Z.Y.; funding acquisition, Z.Y. All authors have read and agreed to the published version of the manuscript.

**Funding:** This research was funded by Shanghai Agriculture Applied Technology Development Program, China (Grant No. 20180107) "Demonstrating Items of Technological Innovation in the Whole Industry Chain of Major Industries: Researches, Integration and Demonstration of Key Technologies for Industrial Upgrading of Fengxian Yellow Peaches". (Funder: Shanghai Agriculture and Rural Affairs Committee).

**Institutional Review Board Statement:** Not applicable.

**Informed Consent Statement:** Not applicable.

**Data Availability Statement:** The data that support the finding of this study are available from the corresponding author upon reasonable request.

**Acknowledgments:** The authors would like to thank Deping Zhou from the Eco-Environmental Protection Institute, Shanghai Academy of Agricultural Sciences, for checking the photographs of TEM samples, and Dapeng Sun from Crop Breeding and Cultivation Research Institute, Shanghai Academy of Agricultural Sciences, for checking the photographs of SEM samples.

**Conflicts of Interest:** The authors declare no conflict of interest.

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
