# Peer review of "Comparison of Waterlogging Tolerance of Three Peach Rootstock Seedlings Based on Physiological, Anatomical and Ultra-Structural Changes"

_horticulturae, doi:10.3390/horticulturae8080720_

Round 1
Reviewer 1 Report
It is a great honnor to read a paper like that. I congrats all the authors. I'm at your disposal to contributed to futere works and colaborations.
In abstract: Written very well. It’s very involving to the readers. But please, double check at keywords. Do not repeat same words in the title. So, I suggest keywords as mycorrhizal fungus infection rates, proline content, superoxide dismutase (SOD) activity per example.
2.1.
It’s very important section, so, I suggest more information’s like photon intensity? Light quality?
2.2. What the tap water composition?
2.9. Where is the M&M to mycorrhizal fungal infection (what software was used to red and yellow asterisk?)
2.10. All test are very chosen. But a correlation between all variables can help you to understand and support all conclusions. Heatmap can be useful. (it is very easy to build using Prism)
Data follow the normality and homogenous variance test? The data were transformed?
3.1.” significant differences”: It is a redundant term. In statistics all differences are significant, do you agree?
3.7.2 Capital letter at the beginning of sentence.
Author Response
Dear Reviewer:
Many thanks for your warm words of support and suggestions for improving the work. The following are our point-by-point answers:
Reviewer 1:
Comments and Suggestions for Authors
- In abstract: Written very well. It’s very involving to the readers. But please, double check at keywords. Do not repeat same words in the title. So, I suggest keywords as mycorrhizal fungus infection rates, proline content, superoxide dismutase (SOD) activity per example.
Answer: The keywords of this manuscript has been amended as follows: waterlogging; peach rootstocks; photosynthetic, antioxidative and osmotic regulation; anatomical adaptation; mycorrhizal fungus infection rates;
- (in 2.1) It’s very important section, so, I suggest more information’s like photon intensity? Light quality?
Answer: Our experiment was conducted in a greenhouse under natural lighting conditions. The temperature settings in the greenhouse were 28 °C/20 °C (±5 °C, day/night), with a relative humidity of about 70 ± 5%. We have made detailed supplementary explanations in the revised manuscript.
- (in 2.2)What the tap water composition?
Answer: To imitate the waterlogging stress in an open field, we conducted our experiment using regular tap water without any extra pre-treatment. The two references below, we hope, may help shed some light on Shanghai's typical tap water's composition.
- Mak Y L, Taniyasu S, Yeung LWY, et al. Perfluorinated Compounds in Tap Water from China and Several Other Countries. Environmental Science & Technology, 2009, 43(13): 4824-9.
- Liu M, Yin H, Wu Q. Occurrence and health risk assessment of pharmaceutical and personal care products (PPCPs) in tap water of Shanghai. Ecotoxicology and Environmental Safety, 2019, 183:
- (in 9). Where is the M&M to mycorrhizal fungal infection (what software was used to red and yellow asterisk?)
Answer: To depict the mycorrhizal fungal infection sites, we used MS Paint and Photoshop to draw the red and yellow asterisk, and blue arrows.
- (In 10). All test are very chosen. But a correlation between all variables can help you to understand and support all conclusions. Heatmap can be useful. (it is very easy to build using Prism) . Data follow the normality and homogenous variance test? The data were transformed?
Answer: After the normality and homogeneity test revealed that the data in the current study were normal and homogeneous, the data were analyzed using the t-test with the aid of SPSS. In the revised version of this manuscript, a heatmap, a membership function analysis, and a principal component analysis were additionally provided to to better explain the correlation between all variables and the contributions for waterlogging performance of rootstock seedlings.
- (in 1).” significant differences”: It is a redundant term. In statistics all differences are significant, do you agree?
Answer: I absolutely agree that all differences in statistics are significant, but in this sentence, "substantial disparities in the change patterns of leaf SPAD value" were to highlight the wider range of change in leaf SPAD value than PH and SD of seedlings.
- (in 7.2)Capital letter at the beginning of sentence.
Answer: We capitalized the initial letter the beginning of sentence of this paragraph: “Waterlogging stress increased the thickness of the upper epidermis of M by 34.1%, while that of S decreased by 29.9% (Table 3). ”

Reviewer 2 Report
the present paper fits within the scope of the Journal as well as the special issue. The topic of the manuscript is quite original since it deals with different rootstocks used in peach to mitigate the negative effect of important abiotic stress such as waterlogging.
I was impressed by the number of measurements in particular the physiological ones.
However, I have some minor comments before the acceptance of the manuscript for publication in Horticulturae:
1) the experimental design is not clear how many replicates and how many plants per experimental unit were used.
2) what about the irrigation scheduling how was performed? more details are required for the readers of Horticulturae.
3) in the discussion section it is not enough to say that your results are in line or not with the scientific literature but why the authors should try to explain why in their own words it is very important.
4) A principal component analysis is required in this case.
5) The conclusion section should be absolutely re-written it seems a duplicate of the results section.
6) I have main concerns about the robustness of the results since the experiment was carried out under a greenhouse experiment and in pots for only one month. What will happen if the trial was conducted under real conditions such as an open field? The authors should absolutely discuss that!
Best of luck
Author Response
Dear reviewer:
Many thanks for your warm words of support and suggestions for improving the work. The following are our point-by-point answers:
Reviewer 2:
Comments and Suggestions for Authors
- the experimental design is not clear how many replicates and how many plants per experimental unit were used.
Answer: In paragraph 2.2 of this manuscript, we stated that the WT lasted 14 days and that a randomized block design was used. Each treatment had four replicates, with six individuals (seedlings) in each. This means that for the following measurements, 4 replicates and 6 plants per experimental unit were used, totaling 24 plants.
- what about the irrigation scheduling how was performed? more details are required for the readers of Horticulturae.
Answer: The waterlogged seedlings do not require irrigation during the overall treatment (14 days). However, the tested seedlings were thoroughly watered once every three days before the waterlogging treatment began.
- in the discussion section it is not enough to say that your results are in line or not with the scientific literature but why the authors should try to explain why in their own words it is very important.
Answer: In the revised version of this manuscript, we re-wrote the part of discussion section and try to explain the differences between our result and the previous conclusions in our own words.
- A principal component analysis is required in this case.
Answer: In the revised version of this manuscript, a heatmap, a membership function analysis, and a principal component analysis were additionally provided to to better explain the correlation between all variables and the contributions for waterlogging performance of rootstock seedlings.
- The conclusion section should be absolutely re-written it seems a duplicate of the results section.
Answer: The conclusion section was re-written in the revised version of this manuscript.
- I have main concerns about the robustness of the results since the experiment was carried out under a greenhouse experiment and in pots for only one month. What will happen if the trial was conducted under real conditions such as an open field? The authors should absolutely discuss that!
Answer: Actually, we have been conducting another batch of waterlogging tests since July 2022 to compare the waterlogging tolerance performance of one-year old seedlings under greenhouse and open field conditions, in order to check the robustness of the results of this study. A similar conclusion was reached, and a new manuscript was in the works.
